# Drug-Disease Severity and Target-Disease Severity Interaction Networks in COVID-19 Patients

**DOI:** 10.3390/pharmaceutics14091828

**Published:** 2022-08-30

**Authors:** Verena Schöning, Felix Hammann

**Affiliations:** Clinical Pharmacology and Toxicology, Department of General Internal Medicine, Inselspital, Bern University Hospital, University of Bern, 3010 Bern, Switzerland

**Keywords:** COVID-19, network analysis, drug-disease interaction, target-disease interaction, DPP4 inhibitors, lipid rafts, drug repurposing

## Abstract

Drug interactions with other drugs are a well-known phenomenon. Similarly, however, pre-existing drug therapy can alter the course of diseases for which it has not been prescribed. We performed network analysis on drugs and their respective targets to investigate whether there are drugs or targets with protective effects in COVID-19, making them candidates for repurposing. These networks of drug-disease interactions (DDSIs) and target-disease interactions (TDSIs) revealed a greater share of patients with diabetes and cardiac co-morbidities in the non-severe cohort treated with dipeptidyl peptidase-4 (DPP4) inhibitors. A possible protective effect of DPP4 inhibitors is also plausible on pathophysiological grounds, and our results support repositioning efforts of DPP4 inhibitors against SARS-CoV-2. At target level, we observed that the target location might have an influence on disease progression. This could potentially be attributed to disruption of functional membrane micro-domains (lipid rafts), which in turn could decrease viral entry and thus disease severity.

## 1. Introduction

In Switzerland, patients seen by general practitioners have a median of two chronic conditions, and receive a median of two prescribed drugs [1]. The most common conditions are cardiovascular diseases, including arterial hypertension and lipid disorders, and diabetes [2]. Not only do drug-drug interactions increase with pill burden, but also the risk for drug-disease interactions (DDSIs), where drugs that are beneficial in one disease may be harmful in another [3]. A drug’s action is brought about by its interaction with molecular targets. The relationship is asymmetric, meaning that a given drug can interact with multiple targets, and one target with multiple drugs [4]. By consequence, the interaction of drugs with specific molecular targets can also influence the progression or severity of a disease, which could lead to a target-disease interaction (TDSI).

The current pandemic of coronavirus disease 2019 (COVID-19) is caused by the severe acute respiratory syndrome coronavirus 2 (SARS-CoV-2). By now, several risk factors for severe COVID-19 progression are known, such as age [5,6], male sex [7,8], or obesity [9,10,11,12]. Additionally, common co-morbidities such as diabetes [13,14,15], cardiac [16,17] and pulmonary diseases [18,19], or dementia [20] can influence prognosis of COVID-19. Furthermore, both the number and the combination of certain co-morbidities have been found to be predictors of severity [21]. Several studies have already been conducted to analyze the influence of specific co-medications on COVID-19 incidence and progression. For example, hypertension is a common chronic condition and a risk factor for severe COVID-19 progression [22]. Some researchers analyzed the influence of anti-hypertensive drugs acting on the renin-angiotensin-aldosterone-system (RAAS)-system [23,24]. The majority of these studies provided evidence that angiotensin converting enzyme (ACE) inhibitors and angiotensin-receptor blockers (ARBs) do not adversely affect the COVID-19 progression or may even be beneficial [22,23,24,25,26,27,28]. In general, studies showed that polypharmacy increases the risk for severe COVID-19 [29,30].

Network analysis is used to investigate a group of objects (e.g., friends, internet servers, patients, enzymes, or proteins) and their connection with each other. The objects are the nodes of the network, whereas the relationships are the edges connecting the nodes. One famous example is Zachary’s “karate club” network, which displays the pattern of friendships amongst the members of a university karate club [31]. In recent years, network analysis has increasingly been applied in the context of pharmacology, e.g., to investigate the relationships between drugs and their respective targets [4] or the relationship between proteins and metabolites [32]. In addition, several network studies on the repurposing of drugs against SARS-CoV-2 have been conducted, mainly as drug-target, target-human, viral-human, or protein-protein-interactions, or combinations thereof [32,33,34]. In addition, transcriptomes of COVID-19 patients, patients with related conditions and healthy controls were compared to identify possible drugs candidates for repurposing [35].

However, none of these studies used clinical data to investigate the influence of pre-existing drug treatment on patient outcomes as measure of disease severity.

The aim of this study was to analyze the impact of DDSIs and TDSIs on COVID-19 severity using network analysis as a tool to inform drug repurposing efforts and increase drug safety. We compared drugs on admission (i.e., drugs patients were taking before or on the day of admission) and their molecular targets in patients who tested positive for SARS-CoV-2, and used severe (required critical care or died) or non-severe outcome (outpatient or never requiring critical care) as endpoint.

## 2. Materials and Methods

### 2.1. Study Population

We carried out this retrospective study at the Insel Hospital Group (IHG), a tertiary hospital network with six locations and about 860,000 patients treated per year, making it the biggest health care provider in Switzerland. The Cantonal Ethics Committee of Bern approved the protocol (2020-00973). We considered all patients who tested positive for SARS-CoV-2 by reverse-transcriptase polymerase chain reaction (RT-PCR) assay on nasopharyngeal swabs at the IHG between 1 February through 16 November 2020—covering the ‘first wave’ and most of the ‘second wave’ of COVID-19 in the region (Figure 1).

For patients with no registered general research consent status, a waiver of consent was granted by the ethics committee. Objection to the general research consent of the IHG was an exclusion criterion for this study, whereas participation in other trials (including COVID-19 related treatment studies) was not. Disease progression was classified as *severe* if, for any reason, an intensive care unit (ICU) admission was required at any stage, or the patient died during the stay. All other patients were classified as *non-severe*. We selected only patients for whom drugs on admission had been recorded. Therefore, this study included 115 severe and 390 non-severe COVID-19 patients. We identified pre-existing conditions using Natural Language Processing from a previous study [36]. For a total of 28 patients (14 non-severe and 14 severe cases), we could not perform disease detection. Characteristics of the study population are provided in Table 1.

To study the effects of co-morbidities, we created four sub-groups:Cardiac conditions (chronic heart failure, atrial fibrillation, coronary heart disease, and/or coronary sclerosis) (*n* = 184)Arterial hypertension (*n* = 246)Diabetes (including pre-diabetes, type 1 and 2 diabetes) (*n* = 139)Dementia (*n* = 54)

Note: patients can be members of more than one group, e.g., 70 patients suffered from diabetes as well as from cardiac conditions.

### 2.2. Network Analysis

Drugs on admission (drugs taken before admission to the IHG) were obtained from the electronic health records (EHR). As this part of the EHR was not always complete, we also considered drugs administered in-house on the day of admission. This also mitigates the effect of patients transferred from other hospitals compared to patients who were initially admitted to the IHG.

We evaluated different levels of detail in drug classification. First, we compared the fourteen main groups of the Anatomical Therapeutic Chemical (ATC) classification system [37]. Then we selected 90 pharmacological, chemical subgroups or substances, which we categorized in 30 therapeutic groups. We identified drugs in the EHR by ATC codes. By and large, the drug groups and subgroups are based on the categorization of the ATC classification, but some minor deviations are present, e.g., acetylsalicylic acid was included as antithrombotic agents, whereas in the ATC code, it is grouped with the analgesics, an uncommon indication in Switzerland. Considering the hyper-thrombotic state of COVID-19 patients [38], we considered its rheological effect to be more important than its analgesic effect. Further information on our grouping is available in the Appendix A. 

Lastly, we analyzed the molecular targets of the drugs on admission. We used DrugBank [39] to map drugs to targets and their target locations.

As the two severity cohorts are imbalanced, we normalized the number of patients for network analysis in each drug (sub-)group by dividing them through the total number of patients in the respective cohort. The obtained value was used as weight in the network analysis.

A network consists of nodes connected by edges. A node’s weight is determined by the number of patients receiving the drug, and an edge’s weight by the number of patients receiving two drugs simultaneously. In our analysis, drugs, drug classes, and targets were represented as nodes and concurrent use or interaction was represented by connecting undirected edges. Therefore, the weights of nodes or edges are both positively correlated with drug use or target engagement. 

### 2.3. Software and Statistical Tests

Data wrangling, analysis, and visualization were performed in GNU R (version 4.0.2, R Foundation for Statistical Computing, http://www.R-project). Statistical significance levels were defined at a *p* value of <0.05, and determined with the Student’s *t*-test for continuous parameters and Chi-square test for categorical parameters using the *stats* package (version 4.0.2). Network analysis was performed using the *igraph* package (Version 1.2.6) [40]. For network visualization, we used Gephi (Version 0.9.2) [41].

## 3. Results

### 3.1. Network Metrics 

The main network metrics are presented in the Appendix A. Main nodes (hubs) and main edges are defined as those with the highest weight, i.e., largest share of patients taking this drug or drug combination. All main nodes and edges are identical between the severity cohorts, except for one edge in the drug subgroups (non-severe: *other analgesics and antipyretics—heparin*; severe: *other analgesics and antipyretics—antibiotics*). The diameter of the network (maximum distance between any two nodes; or the longest shortest path), was in general larger in the severe cohort. Node betweenness centrality (betweenness, indicating how often a node lies on the shortest path between two other nodes) in the non-severe cohort was higher than in the severe cohort (43 and 24 drug subgroups, respectively). More molecular targets had a higher betweenness in the non-severe than in the severe cohort (418 and 124 molecular targets, respectively). In addition, betweenness values in the non-severe cohort were higher (median: 150 vs. 48 and mean: 225 vs. 104, respectively). In Appendix A, we show nodes with the greatest differences in the betweenness between the cohorts.

### 3.2. DDSI Network

There are significant differences (*p* < 0.05) in all three networks (anatomical/pharmacological group, drug group, and drug subgroup) with regards to the drugs (nodes, Table 2) and drug combinations (edges, Table 3) taken on admission. In all nodes and edges with significant differences, the percentage of occurrence was higher in the cohort with severe disease progression unless stated otherwise. As an example, visualization of the anatomical/pharmacological group network is shown in Figure 2.

No differences can be observed in the anatomical/pharmacological group *Alimentary tract and metabolism* or any of the corresponding (sub-)groups between the severity cohorts considering all diseases. However, *anti-hyperglycemics*, specifically *dipeptidyl peptidase-4 (DPP4) inhibitors* and *sodium glucose co-transporter 2 (SGLT2) inhibitors* (only borderline significant, *p* = 0.06), were taken more often by non-severe COVID-19 patients with cardiovascular conditions or cardiovascular conditions and diabetes (see Appendix A). 

In the anatomical/pharmacological group *Blood and blood forming organs*, *anti-hemorrhagics* and *anti-platelet agents* (even though only borderline significant with *p* = 0.095), and within these groups, especially *Vitamin K and other hemostatics* and *acetylsalicylic acid* (only borderline significant, *p* = 0.085), respectively, were significantly different between the severity cohorts. 

In the anatomical/pharmacological group *Cardiovascular system*, which showed no cohort difference, the drug group *diuretics* and *cardiovascular drugs* had a higher percentage in severe COVID-19. In the former group, *loop diuretics* and in the latter, *beta blockers* are significant differences over all patients regardless of co-morbidity. 

Additionally, non-steroidal anti-inflammatory drugs (NSAIDs) were more often taken by patients with non-severe COVID-19, whereas the opposite was true for opioids (but only borderline significant).

Considering all patients, there are differences in combinations of drugs from anatomical/pharmacological group, drug group combinations, and drug subgroup combinations, but the weight of these edges (percentage of patients) is relatively low in most cases (<15%) (Table 3).

However, the disease-specific analysis revealed that the combination of *anti-hyperglycemics* and *anti-coagulants* was more common in non-severe COVID-19 in patients with cardiac conditions or cardiac conditions and diabetes. In the latter cohort, the combination of *anti-hyperglycemics* and *statins* had a higher percentage in non-severe COVID-19 (see Appendix A).

### 3.3. TDSI Network

The main molecular targets and their relative frequency per cohort are shown in the Appendix A. Molecular targets with highly significant (*p* < 0.001) differences are given in the Appendix A. 

Differences in molecular targets can be divided into two groups. The first group comprises targets which interact with only one specific group of drugs, e.g., antithrombotic agents mostly interact with *coagulation factor X*, *P-selection*, and *antithrombin-III*, whereas diuretics may target members of the *solute carrier family 12*. The second group includes targets that cannot be assigned to just one indication or drug group. *Beta adrenergic receptors* are targets for anti-depressants, anti-hypertensives, and anti-arrhythmics. There are over 2690 significantly different edges in the molecular target network. In the Appendix A, we included the 30 most common edges in the network (Appendix A) and the highly significant edges (*p* < 0.001) (Appendix A).

In Figure 3, we present a filtered version of molecular target networks of both severity cohorts, where only nodes with three or more edges are shown. 

The color of the nodes indicates the location of the molecular target within the cell. In the non-severe cohort, more molecular targets are located within the cell membrane, whereas in the severe cohort more targets are located within the cytoplasm.

## 4. Discussion

The network analysis of drugs and their molecular targets revealed differences between the severity cohorts of COVID-19. Except for one edge, the main nodes (hubs) and edges are identical, however the weights were often slightly higher in the severe cohort. This suggests that the most important drugs and drug combinations are the same between the cohorts, but still, slightly more drugs and drug combinations are taken by the severe cohort. This may be indicative of a subpopulation with more co-morbidities. The larger diameter of the severe network indicates that the drugs and drug combinations are more heterogeneous in this cohort. This is supported by the generally lower betweenness of most nodes in this cohort in absolute values, but also in comparison to the non-severe cohort.

However, co-morbidities and co-medications did not always result in a more severe course. Noteworthy here is the higher percentage of patients with cardiac conditions, or cardiac conditions and diabetes, using anti-hyperglycemics, especially DPP4 inhibitors, and to a lower degree SGLT2 inhibitors in the non-severe COVID-19 cohort. These patients had at least two co-morbidities, which are considered risk factors for a severe course [15,42,43], but had a more favorable outcome under these treatment regimens. DPP4 inhibitors have been shown to be reno- and cardio-protective through the suppression of oxidative stress, inflammation, and improvement of endothelial function [44]. Furthermore, there is evidence that SARS-CoV-2, like MERS-CoV (Middle East respiratory syndrome-related coronavirus), also uses the membrane-bound DPP4 enzyme for viral entry. An inhibition of this enzyme is speculated to reduce viral entry and replication [45,46]. In SARS-CoV-2, a functional network analysis revealed that DPP4 is required in viral processes for viral entry and infection. Furthermore, protein-chemical interaction networks revealed important interactions between DPP4 and the DPP4 inhibitor sitagliptin [47]. Additionally, in animal experiments, DPP4 inhibition resulted in a rise of soluble DPP4 [48,49] which could bind to plasma SARS-CoV-2, reducing the amount of virus able to infect cells [50]. Mutations in DPP4 genes, leading to reduced levels for soluble DPP4, were identified as risk factors for increased susceptibility for MERS-CoV [51]. Within an infected cell, sitagliptin inhibited the SARS-CoV-2 papain-like proteases (PLpro) in an in-cell protease assay [52]. Clinical literature on DPP4 inhibitors in COVID-19 is ambiguous; several studies and meta-analyses have showed favorable effects [53,54,55,56], while some have not [57,58,59]. A review of clinical trials with the DPP4 inhibitor sitagliptin found that most studies showed a favorable effect on COVID-19 progression [50]. Several potential modes of action are discussed apart from the above-mentioned decrease in viral entry, increase in soluble DPP4, or inhibition of viral proteases. It is hypothesized that DPP4 inhibitors might attenuate COVID-19-related cardiovascular injury including arrhythmia, acute coronary syndrome and heart failure [60]. In addition, DPP4 inhibition has anti-inflammatory and immunomodulatory properties by decreasing activation of nuclear factor kappa beta (NF-κB) activation and expression of inflammatory cytokines [61,62]. These factors could also influence the progression.

A benefit of SGLT2 inhibitors is supported on pathophysiological grounds. SGLT2 inhibitors have been shown to downregulate systemic and adipose tissue inflammation by decreasing the expression of pro-inflammatory cytokines, lessen oxidative stress, and reduce sympathetic activity [63]. Furthermore, treatment with a SGLT2 inhibitors alleviated myocardial and renal fibrosis in mice [64]. In a large randomized trial with COVID-19 patients, treatment with dapagliflozin, a SGLT2 inhibitors, did not result in a statistically significant risk reduction in organ dysfunction and death, or speedier recovery [65]. 

Considering all patients, regardless of the diagnosed co-morbidities, there are some noteworthy differences in the drugs (nodes of the network, Table 2) used within the cohorts. Despite doubts early in the pandemic regarding the use of NSAIDs during COVID-19 [66], a systematic review and meta-analysis was not able to confirm this theoretical risk [67]. In human cell cultures and mice, NSAIDs reduced pro-inflammatory cytokines, and dampened the humoral immune response to SARS-CoV-2 [68]. This protective effect might be explained by reversing the progressive inflammation in different organs [69]. Even though this study included only few patients on NSAIDs, they were still more common in non-severe patients and thus corroborated earlier studies. Comparisons to other antipyretics with no anti-inflammatory action (e.g., acetaminophen) are necessary.

Some drugs with significant differences between cohorts might be more indicative of the severity of the underlying condition and not interact with COVID-19 prognosis directly. Loop diuretics, for instance are used in more advanced stages of renal failure [70]. As poor renal function is indicative of severe COVID-19 [71,72], this correlation might be due to the severity of the pre-existing condition, not the drug itself. Beta blockers were more often used in the severe cohort, but this might be explained by the higher prevalence of cardiovascular co-morbidities in this cohort. However, loop diuretics, beta blockers, and opioids are also associated with death or severe COVID-19 in a polypharmacy setting [29,30].

Overall, a relatively small percentage of patients received antipsychotic drugs, and the difference between cohorts was not significant (8% and 13% in the non-severe and severe cohorts, respectively). However, combinations with other drugs such as loop diuretics, opioids, beta-blockers, or proton pump inhibitors were more often seen in patients in the severe cohort. The influence of antipsychotic drugs on COVID-19 infection risk and prognosis is currently under discussion. A retrospective study in 698 patients using antipsychotic drugs revealed a lower infection risk and a better prognosis compared to non-users [73]. Comparable results were also reported from a study in patients with a pre-existing diagnosis of schizophrenia, schizoaffective disorder, or bipolar disorder [74]. On the other hand, a systematic review and meta-analysis showed a correlation between antipsychotics and COVID-19 mortality [75]. However, the reviewed studies included patients on antipsychotics independently of diagnoses, considered antipsychotics as a single homogenous pharmacological group, and did not test for adherence [76]. Our results suggest that not the use of a specific drugs per se, but the combination with other drugs influences the risk for severe COVID-19. Therefore, a detailed analysis of the most significantly different drug combinations (edges of the network, Table 3) was performed. Most drug combinations were taken by less than 15% of the patients, which makes a detailed analysis of cause and effect difficult, but trends are visible. In all cases but one (NSAIDs/other analgesics) a greater proportion was seen in the severe cohort. However, this difference is not due to general polypharmacy, which is known to influence disease severity in COVID-19 [29,30], as the number of drugs on admission was not significantly different in both cohorts. Not only polypharmacy, but also specific drug classes influence severity in COVID-19 [29,30]. Drug classes with an increased risk for severe COVID-19 are highlighted in red in Table 3. In seven and in eleven drug combinations, one or both drugs, respectively, were considered high risk. All these combinations were more prevalent in the severe cohort. Only in one combination (NSAIDs/other analgesics), neither drug was considered high risk. Interestingly, a higher proportion of non-severe patients took that combination.

In proton pump inhibitors (PPIs), the effect of combination with other drugs can be seen. PPIs are taken to the same extent by the non-severe and severe cohort (32.6% and 33.9%, respectively, *p* = 0.88, data not shown). However, combinations of PPIs with antipsychotics or platelet inhibitors were more prevalent in severe patients. Several review articles evaluating the effects of PPIs on COVID-19 progression and mortality revealed high heterogeneity in the outcomes [77,78,79,80]. However, those studies did not control for co-medication, except for one which looked at NSAIDs [80]. In summary, studies on drug effects should also consider including and ideally control for co-medication.

In the molecular target network of the non-severe cohort, there are more targets located in the cell membrane. Several hypotheses could help explain this finding. One hypothesis is that interaction of drugs with cell membrane receptors might interfere with viral entry into the cell. The host protein angiotensin-converting enzyme 2 (ACE2) is considered the main entry receptor for SARS-CoV-2 and the transmembrane serine protease 2 (TMPRSS2) an important priming enzyme required during this process [81,82]. In addition, other cell membrane receptors may be involved in cellular entry of SARS-CoV-2 [81,83,84], like neurophilin-I [85,86], or DPP4 [45,46]. Interference may be direct if a drug targets a protein, which is also important for viral entry. Studies on SARS-CoV-2–human protein-protein interaction revealed hundreds of further possible targets [87,88,89,90], however there is only minimal overlap with the target we identified. However, interference may also be indirect due to changes in membrane organization that negatively impact any part of the viral replication cycle. Functionally organized micro-domains (lipid rafts), characterized by highly ordered and tightly packed lipid molecules, within the cell membrane may play a pivotal role in different processes during the viral life-cycle, including coronaviruses [91]. Lipid raft involvement in viral entry was already shown for the murine hepatitis virus, a betacoronavirus such as SARS-CoV-2 [92]. A further study used SARS-CoV-2 pseudo viruses to demonstrate the importance of cholesterol-rich membrane lipid raft for infection [93]. Micro-domains may increase the efficiency of infection by clustering enzymes and receptors in certain membrane area, thus allowing multivalent binding of virus particles, but are not an absolute requirement for the entry process [94]. Several drugs acting on specific the cell membrane targets were shown to disrupt lipid rafts [95]. These included targets we identified in the non-severe cohort, such as alpha- and beta-adrenergic receptors, and opioid receptors. As the network visualizations only include nodes with three or more edges, one might conclude that the combination of several drugs, which interfere with the integrity of the lipid rafts, have an influence on COVID-19 progression.

Our study has some limitations. The severity cohorts had some significant differences in demographics and co-morbidities. The severe cohort was significantly older, had a higher BMI, and a higher share of male patients, all factors which are known risk factors for severe COVID-19 [5,6,7,8,9,10,11,12]. Even though the differences are significant, they are still rather small (median age difference three years, median BMI difference 1.68 points), so that a detailed analysis of these factors would require a larger sample size to obtain enough power with an unknown effect size. Additionally, more patients in the severe cohort suffered from arterial hypertension, chronic heart failure and/or coronary heart disease, again established risk factors in COVID-19 [16,17,22]. However, the presence and even the severity of co-morbidities were indirectly accounted for by the analysis of prescribed drugs. The number of drugs on admission was not significantly different between the cohorts. Even though we were able to include a total of 505 patients in our analysis, the number of patients receiving one specific drug was still relatively low, especially in the disease cohorts. Therefore, significant differences were in some cases only seen in the high-level pooled groups. For this reason, we also reported borderline significant results (0.05 < *p* < 0.1), which could be interpreted as a weak signal and should be investigated in further research. Furthermore, we were mainly able to consider hospitalized patients because of data availability issues. As the IHG is an important regional medical center, some patients were transferred from smaller hospitals. Drugs given in these smaller hospitals are recorded on the drugs on admission list.

The analysis only focused on dual combinations. While we did perform cluster analyses to find more complex combinations, the data available did not support this. Furthermore, even though there were significant differences between the severity cohort with regards to age and sex, we did not control for that.

## 5. Conclusions

In summary, the use of a network approach allowed for studying the impact of drugs from a novel vantage point. Most importantly, autonomic targets appear to be influential on the course of disease in COVID-19, mostly in the form of off-target effects, possibly by disrupting lipid rafts and impeding viral entry. This also holds for DPP4 inhibitors, which are known to interact with adrenergic receptors [96]. The impact of interference with autonomic receptors merits further study into potential future treatments for infection with SARS-CoV-2 and other viruses. Overall, our network analysis indicates that DPP4 inhibitors are related to a better prognosis for COVID-19 and thus represent potential repositioning drugs against SARS-CoV-2. Additionally, our study revealed (i) that drug-induced changes in cell membrane architecture might influence disease progression and (ii) that the influence of specific drugs on disease progression might be dependent on concurrent co-medication.

## Figures and Tables

**Figure 1 pharmaceutics-14-01828-f001:**
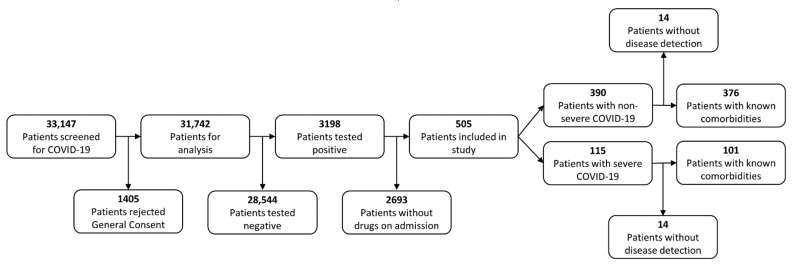
Flowchart of patient selection process.

**Figure 2 pharmaceutics-14-01828-f002:**
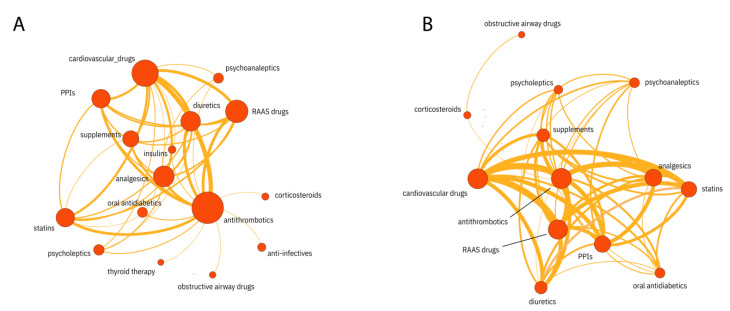
Drug group networks for severe (**A**) and non-severe (**B**) COVID-19 (only nodes with three or more edges are shown).

**Figure 3 pharmaceutics-14-01828-f003:**
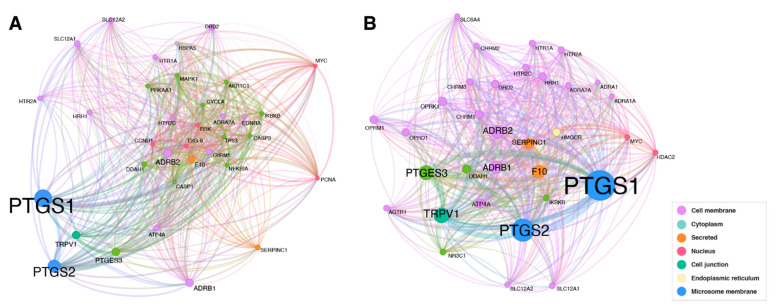
Molecular target networks for severe (**A**) and non-severe (**B**) COVID-19 (only nodes with three or more edges are shown); *ADRA1A/2A*: Alpha-1A/2A adrenergic receptor; *ADRB1/2*: Beta-1/2 adrenergic receptor; *AGTR1*: Type-1 angiotensin II receptor; *AKR1C1*: Aldo-keto reductase family 1 member C1; *ATP4A*: Potassium-transporting ATPase alpha chain 1; *CASP1/3*: Caspase-1/3; *CCND1*: G1/S-specific cyclin-D1; *CHRM1/M2/M3*: Muscarinic acetylcholine receptor M1/M2/M3; *CYCLA*: Cyclin A; *DDAH1*: N(G),N(G)-dimethylarginine dimethylaminohydrolase 1; *DRD2*: Dopamine D2 receptor; *EDNRA*: Endothelin-1 receptor; F10: Coagulation factor X; *HDAC2*: Histone deacetylase 2; *HMGCR*: 3-hydroxy-3-methylglutaryl-coenzyme A reductase; *HRH1*: Histamine H1 receptor; *HSPA5*: 78 kDa glucose-regulated protein; *HTR1A/2A/2C*: 5-hydroxytryptamine receptor 1A/2A/2C; *IKBKB*: Inhibitor of nuclear factor kappa-B kinase subunit beta; *MAPK1*: Mitogen-activated protein kinase 1; MYC: Myc proto-oncogene protein; *NFKBIA*: NF-kappa-B inhibitor alpha; *NR3C1*: Glucocorticoid receptor; *OPRD1*: Delta-type opioid receptor; *OPRK1*: Kappa-type opioid receptor; *OPRM1*: Mu-type opioid receptor; *PCNA*: Proliferating cell nuclear antigen; *PRKAA1*: 5’-AMP-activated protein kinase; *PTGES3*: Prostaglandin E synthase 3; *PTGS1/2*: Prostaglandin G/H synthase 1/2; *RSK*: Ribosomal protein S6 kinase alpha-3; *SERPINC1*: Antithrombin-III; *SLC12A1/2*: Solute carrier family 12 member 1/2; *SLC6A4*: Sodium-dependent serotonin transporter; *TP53*: Cellular tumor antigen p53; *TRPV1*: Transient receptor potential cation channel subfamily V member 1; *TSG-6*: Tumor necrosis factor-inducible gene 6 protein.

**Table 1 pharmaceutics-14-01828-t001:** Characteristics of study population.

General Characteristics	Non-Severe (*n* = 390)	Severe (*n* = 115)	*p* Value
Age (years)			
Median (Q1, Q3)	67.00 (52.00, 77.00)	70.00 (60.50, 81.00)	<0.001
Sex			
Female (%)	155 (39.74%)	31 (26.96%)	0.017
BMI			
Median (Q1, Q3)	26.05 (23.51, 29.43)	27.73 (24.74, 31.70)	<0.006
Drugs on admission			
Median (Q1, Q3)	7.00 (4.00, 12.00)	8.00 (4.00, 13.00)	0.403
**Diseases**			
Arterial hypertension (%)	182 (48.40%)	64 (63.37%)	0.011
Chronic heart failure (%)	92 (24.47%)	37 (36.63%)	0.021
Atrial fibrillation (%)	57 (15.16%)	23 (22.77%)	0.095
Coronary heart disease (%)	52 (13.83%)	32 (31.68%)	<0.002
Coronary sclerosis (%)	9 (2.39%)	6 (5.94%)	0.136
Diabetes (%)	105 (27.93%)	34 (33.66%)	0.316
Dementia (%)	39 (10.37%)	15 (14.85%)	0.278

**Table 2 pharmaceutics-14-01828-t002:** Significant nodes of the DDSI network.

Anatomical/Pharmacological Group	Non-Severe COVID-19 [%]	Severe COVID-19 [%]	*p* Value
Blood and blood forming organs	85.64	94.78	0.014
Various	4.1	10.43	0.018
Musculo-skeletal system	21.79	13.91	0.085
**Drug Groups**			
Anti-hemorrhagics	0.51	3.48	0.037
Diuretics	23.08	32.17	0.064
Cardiovascular drugs	36.67	46.09	0.087
Antiplatelet agents	23.08	31.3	0.095
**Drug Subgroups**			
NSAID	12.56	4.35	0.020
Loop diuretics ^1^	14.87	24.35	0.025
Beta blockers ^1^	26.41	37.39	0.030
Vitamin K and other hemostatics	0.51	3.48	0.037
Opioids ^1^	10.51	17.39	0.068
Acetylsalicylic acid	21.28	29.57	0.085

^1^ Drug subgroups that were associated with death or severe COVID-19 by Iloanusi et al. and McKeigue et al. [29,30].

**Table 3 pharmaceutics-14-01828-t003:** Significant edges of the DDSI network.

Anatomical/Pharmacological Group Combinations		Non-Severe COVID-19 [%]	Severe COVID-19 [%]	*p* Value
Various	Alimentary tract and metabolism	3.85	10.43	0.012
	Nervous system	3.33	9.57	0.012
	Blood and blood forming organs	3.85	9.57	0.028
**Drug Group Combinations**		**Non-Severe COVID-19 [%]**	**Severe COVID-19 [%]**	***p* Value**
Psycholeptics	Anti-hemorrhagics	0.26	3.48	0.011
Antiplatelet agents	Anti-infectives	7.44	15.65	0.013
Cardiovascular drugs	Diuretics	16.92	26.09	0.004
	Obstructive airway drugs	5.9	12.17	0.039
**Drug Subgroup Combinations**		**Non-Severe COVID-19 [%]**	**Severe COVID-19 [%]**	***p* Value**
Antipsychotics ^1^	Loop diuretics ^1^	1.28	7.83	<0.001
	Opioids ^1^	1.03	6.09	0.004
	Adrenergic inhalants	0.51	4.35	0.008
	Beta blockers ^1^	2.82	8.7	0.012
	Proton pump inhibitors ^1^	3.59	8.7	0.044
	Other analgesics	5.38	11.30	0.044
Heparin ^1^	Direct Xa inhibitors ^1^	0.51	4.35	0.008
	Loop diuretics ^1^	4.62	10.43	0.036
Platelet inhibitors ^1^	Antibiotics	7.44	14.78	0.026
	Loop diuretics ^1^	5.13	11.3	0.032
	Beta blockers ^1^	11.03	19.13	0.034
	Proton pump inhibitors ^1^	10.00	17.39	0.045
	Potassium spare diuretics ^1^	1.03	4.35	0.049
Potassium spare diuretics ^1^	Acetylsalicylic acid	0.77	4.35	0.023
	Adrenergic inhalatives	0.51	3.48	0.037
Loop diuretics ^1^	Opioids ^1^	2.82	7.83	0.032
Vitamin K antagonists ^1^	Thyroid	0.51	3.48	0.037
NSAID	Other analgesics	10.26	3.48	0.038
Beta blockers ^1^	Acetylsalicylic acid	10.77	18.26	0.048

^1^ Drug subgroups that were associated with death or severe COVID-19 by Iloanusi et al. and McKeigue et al. [29,30].

## Data Availability

The source code is available on GitHub: https://github.com/cptbern/Covid19-network-analysis.

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
