# Peer review of "Drug-Disease Severity and Target-Disease Severity Interaction Networks in COVID-19 Patients"

_pharmaceutics, 2022, doi:10.3390/pharmaceutics14091828_

Round 1

Reviewer 1 Report

This article is of great interest to clinical pharmacists. In addition, it contains many elements of innovation and also uses mathematical modeling. The main article lacks the given dietary supplements (the most important), the diets that the patients received, it is a pity that it is presented in the supplement. I am merely proposing that the authors write instead of aspirin, acetylsalicylic acid.

Author Response

Point by point reply

Reviewer 1:

This article is of great interest to clinical pharmacists. In addition, it contains many elements of innovation and also uses mathematical modeling. The main article lacks the given dietary supplements (the most important), the diets that the patients received, it is a pity that it is presented in the supplement. I am merely proposing that the authors write instead of aspirin, acetylsalicylic acid.

Thank you for your very positive and valuable feedback. Regarding the dietary supplements (e.g. vitamins), we did include them in our analysis, but no significant differences could be seen. We cannot exclude the underreporting of the patients' supplements, as patients might consider this information as not relevant (in comparison to prescription drugs) and omit to mention them on admission. Our records do not systematically indicate if the patients follow any specific diets at home (vegetarian, vegan).

We also agree with your notion that aspirin should be called acetylsalicylic acid and changed it accordingly in the manuscript.

Reviewer 2 Report

The manuscript (ID: pharmaceutics-1750333) entitled “Drug-disease severity and target-disease severity interaction networks in COVID-19 patients” reports the analysis of drug-disease and target-disease interactions data collected from patients hospitalized for COVID-19, in search of potential drug repositioning opportunities. The results reported in this study could provide information potentially useful for the repositioning of drugs against SARS-CoV-2 and to improve safety in patients affecting COVID-19. However, there are several issues in the manuscript that should be addressed to be considered for publication.

·        Several factors as age and presence of co-morbidities have been associated to poorer prognosis in COVID-19 patients according to literature data, some of them being also cited in the introduction section of the manuscript. A more detailed discussion of these factors with respect to the observed results should be provided in my opinion. This, especially considering that there might emerge significant differences between the severity cohorts in the reported results.

·        The study took into consideration records related to COVID-19 patients hospitalized between February 1st, 2020 and November 16th, 2020, and then different filtering criteria were adopted obtaining data related to 505 of them. I would suggest reporting the number of records that were retained at each step of the filtering process, to provide valuable, additional details on the adopted protocol. Additional details on the parameters and settings employed, e.g., in the network analysis should also be provided in the Methods section of the manuscript.

·        Data on molecular targets reported in DrugBank was associated to the drugs investigated in the study and the resulting information was analyzed. Several databases specifically associating information focused on targets, diseases, pathways to their molecules have been developed so far. Moreover, public databases of molecules reporting activity data on sets of targets significantly larger than that of DrugBank are also available. Very often, the use of a single database in this type of searches can limit the number and the information on the targets hit by the drugs under investigation, thus restricting the potentially discoverable drug-target-disease interactions. In my opinion, the selection of DrugBank for this task of the analyses should be commented.

·        Some differences between the percentages of the drug combinations in the severe and non-severe cases of COVID-19 can be observed according to Table 3 contents. Most of the higher values of percentages can be observed for the severe cases. A discussion on these results (for each specific case) should be provided, especially considering that patients with co-morbidities are generally likely to need more therapeutic treatments and, at the same time, have higher probabilities to present severe cases of COVID-19 according to literature data. Moreover, a comment on the fact that the “NSAID/Other analgesics” combination presents a different behavior on the percentage values, with respect to the others in Table 3 should also be provided.

·        I would suggest adding a more detailed discussion (and a comparison) on the results and data reported in the study, with respect to information from literature (whether possible, also related to non-hospitalized patients). This could help further strengthening the reported results, while supporting the potential repositioning of drugs, e.g. that inhibit DDP4, against SARS-CoV-2.

·        A more detailed description on the contents of Tables S2 to S4 should be provided in their respective captions. Moreover, tables and figures reported in the Supporting Information should properly numbered in the Main Section. For example, Table S4 appears before Tables S2 and S3 within the manuscript.

·        The acronyms should be spelled out at their first occurrence in the main section of the manuscript to avoid potential misunderstandings (some examples: ICU at page 2; HER at page 3; NSAID at page 5).

In addition to the previous comments, the manuscript is sometimes not easily readable, with typos in some parts. I would suggest a revision of the language. Moreover, the bibliography is sometimes not exhaustive in my opinion. For example, the sentences ending at lines 45, 47 and 48 (pages 2), 237 (page 7), and 241, 249 and 269 (page 8) would need to be properly referenced.

Author Response

Dear Reviewer 2,

please find point by point reply at the attachment

Reviewer 3 Report

Dear Authors,

I have read the manuscript and I sedn you my comments:

1) I am not sure that the journal is correct, I think that it is a epidemiological or informatic study more than a study on drug use or pharmacodynamic effects.

2) I hev not understand the type of study, usually this method is used to explaine a phenomena or a mechanism.

3) No clinical data have been described

4) The method is used to evaluate on documents more than on humans the effects, so why you have enrolled patients?

5) there is not correlation with dosage, kydney function age gender or other parameters.

6) no data on ethic committe have been described

Author Response

Dear Reviewer 3,

Please find point by point reply at the attachment

Reviewer 4 Report

This is a study showing the potential for COVID-19 treatment of previously approved DPP4 inhibitors through network analysis. I have some comments that I believe might help the authors in increasing the impact of this manuscript.

1. DPP4 is incorrectly described as DDP4.

2. In addition to the discussion on Figure 2, additional explanations related to the action of the DPP4 inhibitor and the DDSI network analysis are needed.

3. Check the abbreviation. e.g., ICU, EHR etc.

4. Line 244-246: It is necessary to update the recently published research papers.

5. Increase the font size of the legend display according to the color of Figure 2. It is difficult to check the contents.

6. References: Please check reference style. In particular, journal abbreviation names.

Author Response

Dear Reviewer 4,

Please find point by point reply at the attachment

Round 2

Reviewer 2 Report

The revised version of the manuscript “pharmaceutics-1750333” presents substantial improvements. At the present state, I can recommend the manuscript for publication on the Journal.

Reviewer 3 Report

Dear Authors,

I have read the manuscript and it has actually been improved. However, no clinical and pharmacological data are reported. No data on dosage or type of drug in each people is described. Furthermore, these data are likely similar to the data in patients without COVID-19.